# Cooperative supramolecular polymers with anthracene–endoperoxide photo-switching for fluorescent anti-counterfeiting

Zhao Gao[1], Yifei Han[1] & Feng Wang [1]

Innovative technologies are highly pursued for the detection and avoidance of counterfeiting in modern information society. Herein, we report the construction of photo-responsive supramolecular polymers toward fluorescent anti-counterfeit applications, by taking advantage of multicycle anthracene–endoperoxide switching properties. Due to $\sigma$-metalation effect, photo-oxygenation of anthracene to endoperoxide is proceeded under the mild visible light irradiation conditions, while the backward conversion occurs spontaneously at room temperature. Supramolecular polymers are formed with cooperative nucleation–elongation mechanism, which facilitate fluorescence resonance energy transfer process via two-component co-assembly strategy. Fluorescence resonance energy transfer efficiency is delicately regulated by either light-triggered anthracene–endoperoxide conversion or vapor-induced monomer–polymer transition, leading to high-contrast fluorescent changes among three different states. On this basis, dual-mode anti-counterfeiting patterns have been successfully fabricated via inkjet printing techniques. Hence, cooperative supramolecular polymerization of photo-fluorochromic molecules represents an efficient approach toward high-performance anti-counterfeit materials with enhanced security reliability, fast response, and ease of operation.

[1] CAS Key Laboratory of Soft Matter Chemistry, iChEM (Collaborative Innovation Center of Chemistry for Energy Materials), Department of Polymer Science and Engineering, University of Science and Technology of China, Hefei, Anhui 230026, China. Correspondence and requests for materials should be addressed to F.W. (email: drfwang@ustc.edu.cn)

Counterfeiting, widely spread in banknotes, diplomas, tax stamps, and certificates, has become a serious and social-threatening problem[1]. It is in keen pursuit to develop innovative anti-counterfeiting technologies, which avoids the forgeries and ensures security reliability for the genuine documents[2–5]. In this respect, photo-fluorochromic compounds are regarded as an intriguing choice, because of their reversible emission outputs in response to light[6–8]. Generally, inkjet printable anti-counterfeiting materials are prepared by direct mixing of photo-fluorochromic compounds with the polymer matrix[9,10]. However, their limited inclusiveness potentially leads to the decrease in ink's quality. In addition, most of the reported anti-counterfeiting inks are operated in a single-mode fashion[11–17], which can still be easily replicated. Hence, it is highly desirable to fabricate anti-counterfeiting inks with bright fluorescence and multi-mode responsiveness.

To attain this objective, a feasible approach is to graft photo-fluorochromic molecules onto polymer backbone[11,12]. An alternative strategy is to assemble such photo-responsive units into uniform supramolecular polymers[18–20], which is superior to the covalent approach due to the avoidance of tedious synthesis. Moreover, supramolecular polymers serve as a dynamic platform for multi-component organization[21–27], allowing for the cascade light-induced energy/electron transfer process[28–36]. As a consequence, the fluorescent signals can be fine-tuned in a multi-mode manner.

Based on these considerations, we sought to develop photo-fluorochromic supramolecular polymers toward anti-counterfeiting applications. The molecular design relies on anthracene–endoperoxide interconversion, which displays on/off switching for the emission signals[37–41]. Notably, harsh reaction conditions are required for the traditional anthracene–endoperoxide photo-switching (365 nm UV light irradiation for oxygenation process, while 254 nm UV light or thermolysis for deoxygenation reaction)[37]. Photo-degradation side-products are prone to form during the processes, which hamper the practical application in fluorescent anti-counterfeiting inks. To overcome this issue, herein anthracene-based monomer 1 (Fig. 1) has been designed, with the attachment of Pt(PEt$_3$)$_2$ moiety on both ends of 9,10-diacetylide anthracene. Due to $\sigma$-metalation effect[42–45], photo-oxygenation of 1 can be rapidly proceeded via the less destructive visible light source, while the backward conversion occurs at room temperature in a spontaneous manner. These mild conditions facilitate the quantitative anthracene–endoperoxide switching for multiple cycles. Additionally, two amide units are connected to the rod anthracene core of 1. Ascribed to the synergistic participation of hydrogen bonding and π–π stacking interactions, supramolecular polymerization of 1 is expected to feature distinctive nucleation and elongation events[46–52]. It thus mimics the cooperative process adopted by biological self-assembled systems[53,54], leading to macroscopic fluorescent variations before and after light irradiation. On this basis, a two-component supramolecular system 1/2 is established, which gives rise to fluorescence resonance energy transfer (FRET) from donor 1 to acceptor 2 (Fig. 1). The FRET efficiency can be precisely regulated, by either light-triggered anthracene–endoperoxide or vapor-triggered monomer–polymer transitions. Accordingly, dual-mode anti-counterfeiting pattern (Fig. 1) in response to the orthogonal stimuli is achieved, which guarantees higher security reliability.

## Results

**Spectroscopic characterization of** 1. As a first step, spectroscopic experiments were performed for the designed monomer 1. Specifically, it displays three main UV–Vis absorption bands in CHCl$_3$ (Fig. 2a). According to the previous literatures[55–57], the intense vibronic band at 250–310 nm, together with the moderately intense band at 400–525 nm ($\lambda_{max}$ = 454 and 484 nm for the two absorption maxima), are assigned to $^1L_b$ and $^1L_a$ bands of anthracene unit, respectively. In addition, the broad and intense band locating at 310–395 nm ($\lambda_{max}$ = 331 nm) belongs to π–π* transition of platinum acetylide (–C≡C–Pt–C≡C–) segment, which is absent for the counterpart compound 4 (see the structure in Fig. 2a, inset). Meanwhile, 1 shows the $^1L_a$ emission band centered at 512 nm, along with a vibronic peak at 542 nm

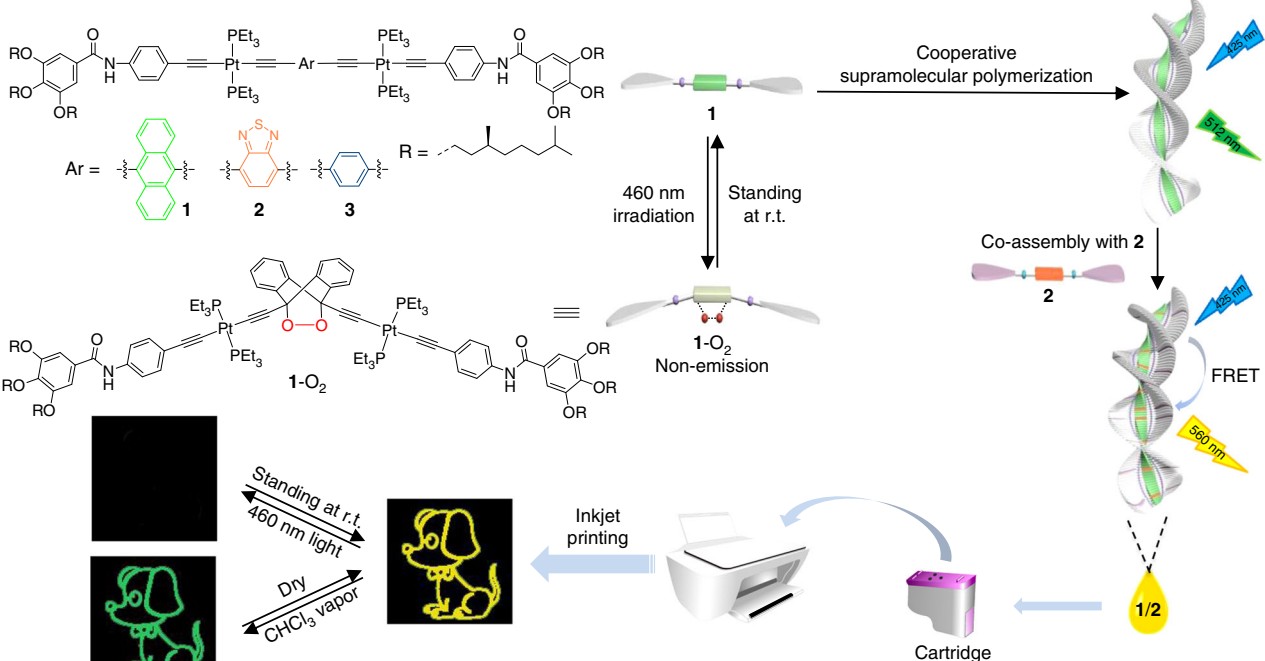

**Fig. 1** Molecular structures design and anti-counterfeiting applications. Schematic illustration of monomer 1 with anthracene–endoperoxide photoswitching, and cartoon representation for the resulting supramolecular polymers toward fluorescent anti-counterfeiting applications

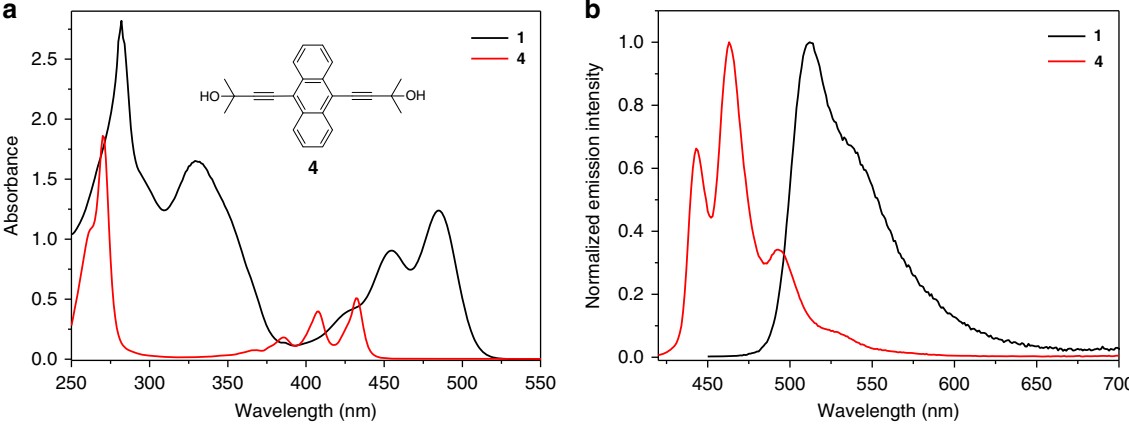

**Fig. 2** Spectroscopic characterization of **1**. **a** UV–Vis and **b** fluorescence spectra of **1** and **4** in CHCl$_3$ ($2.00 \times 10^{-4}$ M, $\lambda_{ex} = 425$ nm for **1** and 408 nm for **4**)

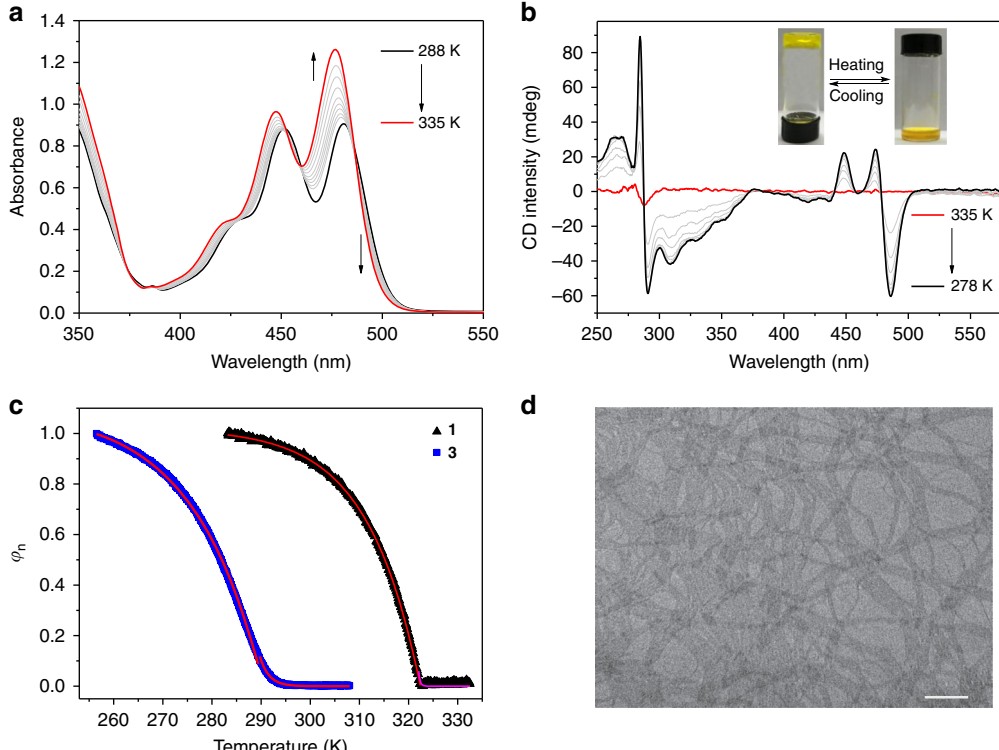

**Fig. 3** Characterization of supramolecular polymers from **1**. **a–b** Temperature-dependent UV–Vis and CD spectra of **1** ($2.00 \times 10^{-4}$ M in MCH). Inset of **b**: thermo-responsive gel–sol transition of **1** ($1.00 \times 10^{-2}$ M in MCH). **c** Net helicity $\varphi_n$ as a function of temperature for **1** ($\lambda = 486$ nm) and **3** ($\lambda = 379$ nm) ($2.00 \times 10^{-4}$ M in MCH). The red and pink lines denote the mathematical fitting. **d** TEM image by drop-casting **1** ($4.00 \times 10^{-4}$ M in MCH) on the copper grid. Scale bar: 500 nm

(Fig. 2b). As compared to **4**, **1** displays *ca*. 50 nm bathochromic shifts for both $^1L_a$ absorption and emission bands (Fig. 2a, b). Such phenomena are primarily ascribed to the overlapping between Pt *d*-orbitals and anthracene *p*-orbitals, resulting in the enhancement of π-electron delocalization[45,55–57].

**Supramolecular polymerization behaviors of 1.** As widely documented, monomeric state dominates for π-conjugated molecules in dilute CHCl$_3$ solution, while self-aggregation is more preferred in aliphatic media[58]. For **1** in methylcyclohexane (MCH), the $^1L_a$ absorbance bands are centered at 451 and 480 nm at 298 K (Fig. 3a). It displays three isosbestic points (374, 460, and 487 nm) upon varying the temperature, suggesting the reversible

transition between self-assembled and monomeric states (Fig. 3a). The strong aggregation tendency of **1** in aliphatic media can be further reflected by $^1$H NMR measurements. Briefly, broadened and unresolved aromatic resonances are present in $d_{12}$-cyclohexane, which are in sharp contrast to the well-defined peaks observed in *d*-chloroform (Supplementary Fig. 2a). Moreover, the two amide units are prone to form intermolecular hydrogen bonds at room temperature, as evidenced by the upfield shifting of amide protons upon elevating the temperature (Supplementary Fig. 2b).

Circular dichroism (CD) spectroscopy is then utilized to probe the regularity of the supramolecular assemblies[59]. To induce the helical bias for **1** at the supramolecular level, optically active (*S*)-3,7-dimethyloctyl groups are introduced to the peripheral side chains (Fig. 1). At 298 K, no CD signals are detected for **1** in

**Table 1 Thermodynamic parameters of 1 and 3 obtained by fitting the temperature-dependent CD data**

| Compound | $T_e$ (K) | $h_e$ (kJ mol$^{-1}$) | $K_a$ |
|---|---|---|---|
| 1 | 322.4 | −97.6 | $2.32 \times 10^{-6}$ |
| 3 | 290.5 | −50.2 | $4.60 \times 10^{-4}$ |

$T_e$: Critical elongation temperature, $h_e$: Enthalpy release upon elongation, $K_a$: Dimensionless equilibrium constant of the activation step at $T_e$. The concentrations for both **1** and **3** are 2.00 × 10$^{-4}$ M in MCH

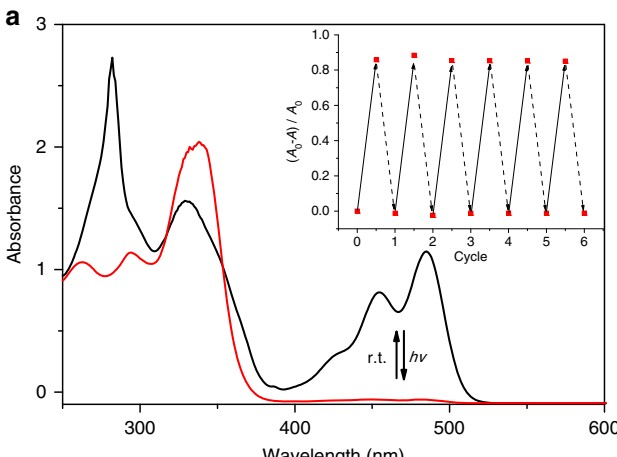

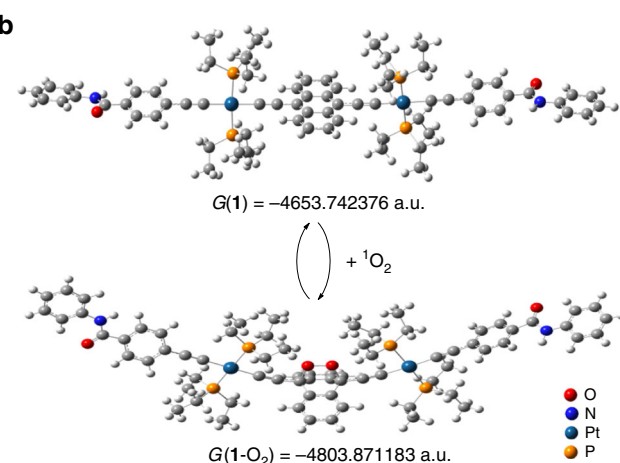

**Fig. 4** Experimental and theoretical studies for photo-switching. **a** UV–Vis spectra change of **1** (2.00 × 10$^{-4}$ M in CHCl$_3$) upon the successive 460 nm light irradiation and standing at room temperature. Inset: multicycle tests by monitoring the absorbance changes at 484 nm. **b** Optimized structures and the Gibbs-free energy of **1** and **1**-O$_2$ on the basis of DFT calculations

CHCl$_3$ (Supplementary Fig. 1), because of the dominance of molecularly dissolved state. In sharp contrast, strong Cotton effects are observed for the $^1L_a$ and $^1L_b$ bands in MCH (for 448 nm, $\Delta\varepsilon = 33.8$ L mol$^{-1}$ cm$^{-1}$, anisotropy factor $g$ value = 0.0008; for 474 nm, $\Delta\varepsilon = 36.8$ L mol$^{-1}$ cm$^{-1}$, $g$ value = 0.00103; for 486 nm, $\Delta\varepsilon = -91.6$ L mol$^{-1}$ cm$^{-1}$, $g$ value = −0.0023, Fig. 3b). Meanwhile, no linear dichroism is detected for **1** (Supplementary Fig. 5), suggesting the measured CD signal is real to reflect the supramolecular chirality[60]. Accordingly, it supports the formation of long-range-ordered supramolecular polymers in apolar MCH medium, accompanying with the chirality transfer from the alkyl periphery to the inner anthracene unit[59].

Supramolecular polymerization mechanism of **1** is further elucidated via temperature-dependent CD experiments. Specifically, CD signals completely lose upon elevating temperature to 322 K, and fully restore upon cooling (Fig. 3b). At the melting rate of 40 K h$^{-1}$, no heating–cooling hysteresis can be observed, revealing that the self-assembly process is under thermodynamic control (Supplementary Fig. 3)[18]. When monitoring the CD intensity at 486 nm versus temperature, a non-sigmoidal melting curve is obtained for **1** (Fig. 3c). Hence, similar to the biological self-assembled polymers such as tobacco mosaic virus and cytoskeleton proteins[53,54], **1** follows cooperative nucleation–elongation mechanism. For the counterpart compound **3** (see the structure in Fig. 1) with the replacement of anthracene unit on **1** by the benzene group, it also shows cooperative supramolecular polymerization behavior (Fig. 3c and Supplementary Fig. 4). Meijer–Schenning–van-der-Schoot mathematical model (Supplementary Equations 1 and 2) is employed to acquire the thermodynamic parameters for both self-assembly processes[46,47]. As shown in Table 1, monomer **1** features the higher $T_e$, $h_e$ and lower $K_a$ values than those of **3**, indicating the formation of more stable polymers with the higher cooperativity[47]. Considering that the structures between **1** and **3** only differ in the inner π-surface, it unambiguously supports that, in addition to intermolecular hydrogen bonds, π–π stacking interactions also participate in the supramolecular polymerization process.

Remarkably, **1** and **3** also display distinct gelation behaviors. In particular, when the monomer concentration of **1** exceeds 8 mM in MCH, it tends to form yellow transparent gels at room temperature (Inset of Fig. 3b). Depending on transmission electron microscopy (TEM) measurements, long ribbon-like fibers are observed for **1**, with several microns in length and around 100 nm in width (Fig. 3d). These one-dimensional fibers are entangled to form fibrous networks, which lay the basis for the formation of supramolecular gels. In sharp contrast, neither gelation (even increasing the concentration to 80 mM) nor entangled long fibers can be visualized for **3** (Supplementary Figs. 6–7). Overall, minor modification on the monomeric structure brings about huge differences in supramolecular polymerization and macroscopic gelation behaviors.

**Anthracene−endoperoxide photo-switching of 1.** We then turned to investigate the photo-responsiveness of **1**. 460 nm LED lamp is employed, considering that **1** absorbs light in this region. Upon photo-irradiation of **1** in CHCl$_3$, both $^1L_a$ absorption (400–525 nm) and emission (475–650 nm) bands decrease rapidly in their intensities, and level off within 20 s (Fig. 4a and Supplementary Fig. 8). Two isosbestic points (318 and 354 nm) are observed in UV–Vis spectra, revealing the conversion from **1** to its photo-chemical product (Supplementary Fig. 8). The process can be further demonstrated by means of $^1$H NMR experiments (Fig. 5a, b), which show the disappearance of anthracene resonances (H$_a$ 8.74 ppm, H$_b$ 7.44 ppm), together with the emergence of a new set of resonances in the upfield region (H$_a$, 7.83 ppm, and H$_b$, 7.34 ppm).

The exact photo-chemical product is then elucidated. When N$_2$ is purged into **1**, both UV–Vis spectroscopic and $^1$H NMR signals hardly change (Supplementary Fig. 9). Accordingly, it suggests the participation of O$_2$ during photo-conversion process. Furthermore, light irradiation is performed for the mixture of **1** and 2,2,6,6-tetramethylpiperidine (TEMP). In electron paramagnetic resonance spectroscopy, three narrow lines are observed with equal intensity ($g$ value = 2.006)[61,62], validating the formation of paramagnetic nitroxide radical, TEMPO (Supplementary Fig. 10). We rationalized that, due to Pt(II) σ-metalation effect,

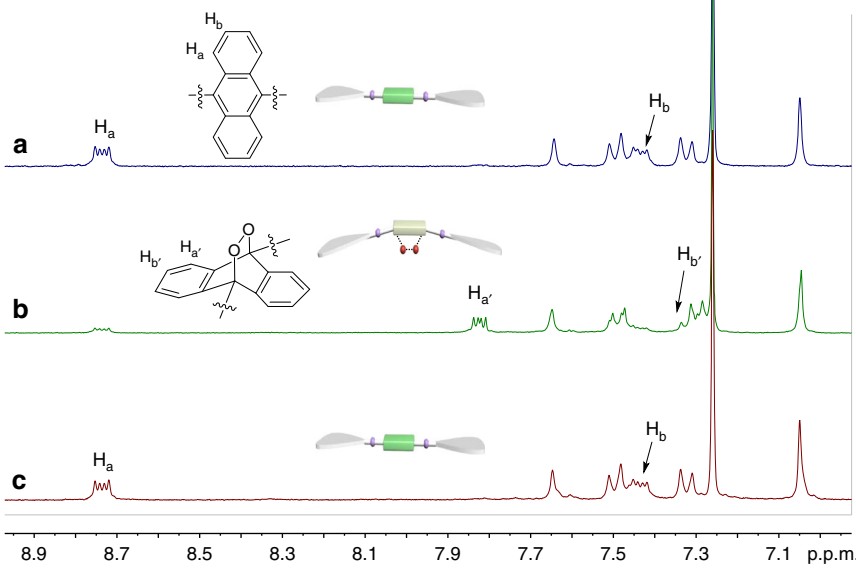

**Fig. 5** Anthracene–endoperoxide photo-switching of **1**. Partial $^1$H NMR spectra (CDCl$_3$, 298 K, 2.00 mM) of **1**: **a** before irradiation, **b** conversion to **1**-O$_2$ after 5 min irradiation, and **c** recovery to **1** upon room temperature standing for 10 min

spin–orbit coupling takes place for **1** via intersystem crossing[43]. It thus facilitates the generation of singlet oxygen ($^1$O$_2$) via energy transfer between the triplet excited state of **1** and the surrounding O$_2$, which is captured by TEMP. Hence, **1** undergoes rapid and quantitative [4 + 2] photo-oxygenation reaction with $^1$O$_2$ to provide the endoperoxide species (**1**-O$_2$, see Fig. 1). Density functional theory (DFT) calculations provide the extra evidence for [4 + 2] photo-transition process (Fig. 4b). In detail, the Gibbs-free energy ($G$) for **1**, **1**-O$_2$, and $^1$O$_2$ are calculated to be –4653.742376 a.u., –4803.871183 a.u., and –150.086995 a.u., respectively. The $\Delta G$ value for photo-oxygenation process is calculated to be –0.0418 a.u. (–109.7853 kJ mol$^{-1}$), demonstrating the thermodynamic favorable conversion from **1** to **1**-O$_2$.

For endoperoxide-to-anthracene transition, deep UV light (254 nm) or thermolysis is commonly required for the conventional 9,10-disubstituted anthracenes[37]. In stark contrast, **1**-O$_2$ converts to **1** at room temperature in a spontaneous manner. The conclusion is verified by the reappearance of anthracene $^1$H NMR resonances (Fig. 5c), together with the complete restore of $^1$L$_a$ and $^1$L$_b$ spectroscopic bands (Supplementary Fig. 11). Half-life time ($t_{1/2}$) for the deoxygenation process is determined to be 15.9 min by UV–Vis measurements ($c = 2.00 \times 10^{-4}$ M for **1**-O$_2$ at 293 K), which can be further modulated upon varying the temperature (3.70 min at 308 K, 8.60 min at 298 K, and 61.0 min at 288 K, see Supplementary Fig. 12). We rationalized that, when comparing to the anthracene unit in **1**, the bulky endoperoxide ring in **1**-O$_2$ imposes larger hindrance to the neighboring two Pt(PEt$_3$)$_2$ units[28]. To release the steric stain, it could undergo spontaneous endoperoxide-to-anthracene transition for **1**-O$_2$ even at room temperature. With the successive light irradiation and room-temperature standing, **1** and **1**-O$_2$ can be reversibly switched for multiple cycles, which are confirmed by the vanishing/reappearance of both absorbance and emission bands (Inset of Fig. 4a, and Supplementary Fig. 11).

We also performed anthracene–endoperoxide photo-switching for the control compound **5** (see the structure in Fig. 6a, inset). Only 60% conversion takes place for the photo-oxygenation process, with the requirement of substantially longer irradiation time (800 s for **5** versus 20 s for **1** at $2.00 \times 10^{-4}$ M, 298 K, see Fig. 6a and Supplementary Fig. 14). For the deoxygenation process (Fig. 6b), the $t_{1/2}$ value for **5**-O$_2$ is determined to be

26 min, which is also slower than that of **1**-O$_2$ (8.6 min). Hence, it proves that Pt(II) $\sigma$-metalation of anthracene is essential for the rapid and quantitative anthracene–endoperoxide switching.

**Photo-responsiveness of supramolecular polymers from 1.** For **1**, anthracene–endoperoxide interconversion occurs not only in the monomeric state (CHCl$_3$ solution), but also in the supramolecular polymeric state (MCH solution). Notably, the kinetics are different between the two states. In particular, the observed rate constant ($k_{obs}$) for photo-oxygenation is 0.0128 s$^{-1}$ in MCH, which is slower than that in CHCl$_3$ ($k_{obs} = 0.71$ s$^{-1}$, see Fig. 6a and Supplementary Fig. 17). Similarity, for the backward conversion from **1**-O$_2$ to **1**, $t_{1/2}$ values vary from 15.9 min in CHCl$_3$ to 34.0 min in MCH (Fig. 6b and Supplementary Fig. 18). The retarded conversion rates in supramolecular polymeric state are probably ascribed to the shielding of photo-fluorochromic units from $^1$O$_2$. Another reason for the different kinetics originates from the varied dipolarity/polarizability for the two solvents[63].

Photo-responsiveness of **1** gives rise to the macroscopic changes of the resulting supramolecular polymers. In detail, the bisignate CD signals of **1** in MCH totally disappear upon 460 nm light irradiation, denoting the loss of supramolecular chirality (Fig. 7a). Besides, supramolecular gels of **1** (10 mM in MCH) completely collapse within 4 min, accompanying with the weakening of green emission (Inset of Fig. 6b, and Supplementary Fig. 19). Upon standing at room temperature, highly emissive supramolecular gels reform, suggesting the complete deoxygenation of **1**-O$_2$.

We then sought to exploit anti-counterfeiting applications of **1**, by taking full advantage of its fluorescent on–off switching properties. Specifically, the supramolecular polymeric solution of **1** is filled in the customized cartridge of an inkjet printer. A monochromic QR code is printed on a non-fluorescent paper, which is invisible under natural light and can be decoded under UV light by a smartphone (Fig. 8a and Supplementary Fig. 21). When the printed paper is exposed to 460 nm light, the QR code is completely erased within 2 min. The initial state restores after standing at room temperature for 10 min, which is accelerated by heat treatment (3 min upon heating at 40 °C). Thanks to the quantitative anthracene–endoperoxide photo-switching of **1**, the

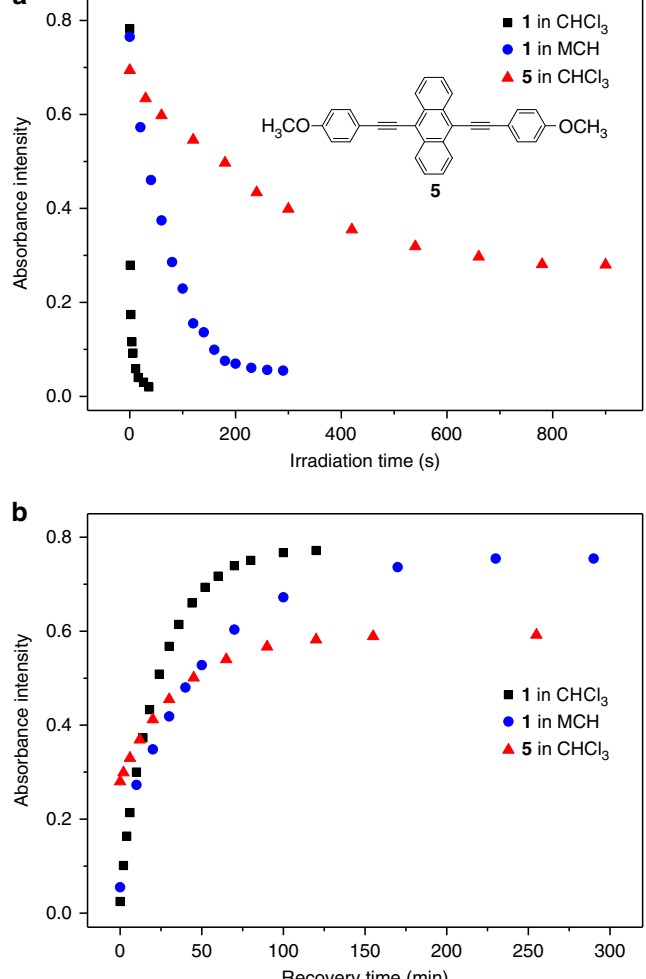

**Fig. 6** Anthracene–endoperoxide photo-switching kinetics. **a** Photo-oxygenation and **b** deoxygenation kinetics of **1** in CHCl₃ (black square), in MCH (blue circle), and **5** in CHCl₃ (red triangle) on the basis of UV–Vis measurements. For **1** and **5** ($2.00 \times 10^{-4}$ M, 293 K), the absorbance signals are monitored at 484 nm, and 470 nm, respectively

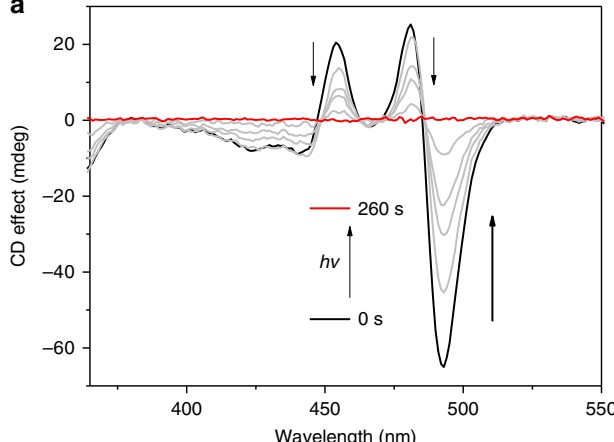

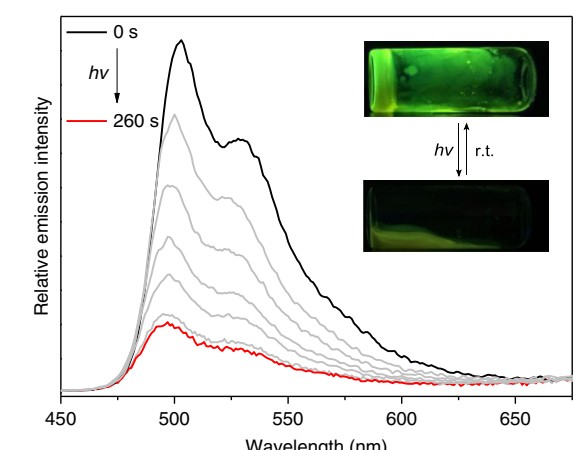

**Fig. 7** Photo-switching of supramolecular polymers from **1**. **a** CD and **b** fluorescent spectra of **1** ($2.00 \times 10^{-4}$ M in MCH at 298 K) upon 460 nm light irradiation for 260 s. Inset of **b**: gel–sol transition of **1** (10.0 mM in MCH) upon the successive 460 nm light irradiation and standing at room temperature

inkjet-printed QR pattern can be read and hidden for multiple cycles (Fig. 8b).

**Dual-mode anti-counterfeiting inks from copolymers** 1/2. On this basis, FRET is introduced to the supramolecular polymeric system via a two-component co-assembly strategy (mixing monomers **1** and **2** together, see Fig. 1), which represents an effective route for the fabrication of multi-mode fluorescent anti-counterfeiting inks. For the efficient FRET process between **1** and **2**, the prerequisite is their close distance and appropriate spectral overlap. Depending on the spectroscopic measurements, **2** features absorbance at 425–575 nm[52], which overlaps well with the emission band of **1** (spectral overlap integral: $J = 9.47 \times 10^{11}$ M⁻¹ cm⁻¹ nm⁴, see Supplementary Figs. 22–23). Hence, monomers **1**–**2** are capable of serving as FRET donor–acceptor pairs.

The FRET efficiency is then evaluated, by progressive addition of **2** into the supramolecular polymeric solution of **1** (molar ratios of **1/2**: from 500: 1 to 10:1). As shown in Fig. 9a, the emission signal of **1** ($\lambda = 505$ nm) decreases in its intensity, while that of **2** centered at 560 nm gradually increases, with an iso-emissive point at 541 nm. During the titration processes, the emission colors change from green to yellow (Inset of Fig. 9a, and Supplementary

Fig. 24). For a 10:1 mixture of **1** and **2** in MCH, $\Phi_{ET}$ (energy transfer efficiency) and $k_{ET}$ (energy transfer rate constant) are determined to be 80% and $2.40 \times 10^{9}$ s⁻¹, respectively. Notably, neither emission spectra nor color changes can be observed for the **1/2** mixture in CHCl₃ (Supplementary Fig. 25), suggesting that FRET occurs only in the co-assembled state.

A dog pattern with yellow emission is then obtained, by inkjet printing co-assembly **1/2** on the non-fluorescent papers (Fig. 9c and Supplementary Fig. 30). Upon 460 nm light irradiation, the emission signal is significantly weakened for its intensity (since **1**-O₂ is non-emissive, the weak fluorescence is derived from the trace amount of **2**), while restored within 3 min at 40 °C. Considering that no chemical transformation takes place for **2** during the light irradiation processes (Supplementary Fig. 29), the fluorescent switching of **1/2** is unambiguously attributed to the interconversion between **1** and **1**-O₂, which deactivate/reactivate the FRET effect (Fig. 9b). In the meantime, fuming the dog pattern with CHCl₃ vapor leads to the appearance of green emission, which promptly turns back to yellow upon drying (Fig. 9c). The phenomena are highly plausible, since polar solvent destroys the co-assembled structure and thereby shows the characteristic fluorescent signal of **1**. More importantly, the anti-counterfeiting fluorescent inks derived from **1/2** feature sufficient stability and sustainability, since almost no emission decay can be visualized upon placing the printed dog pattern under the ambient conditions for 2 weeks

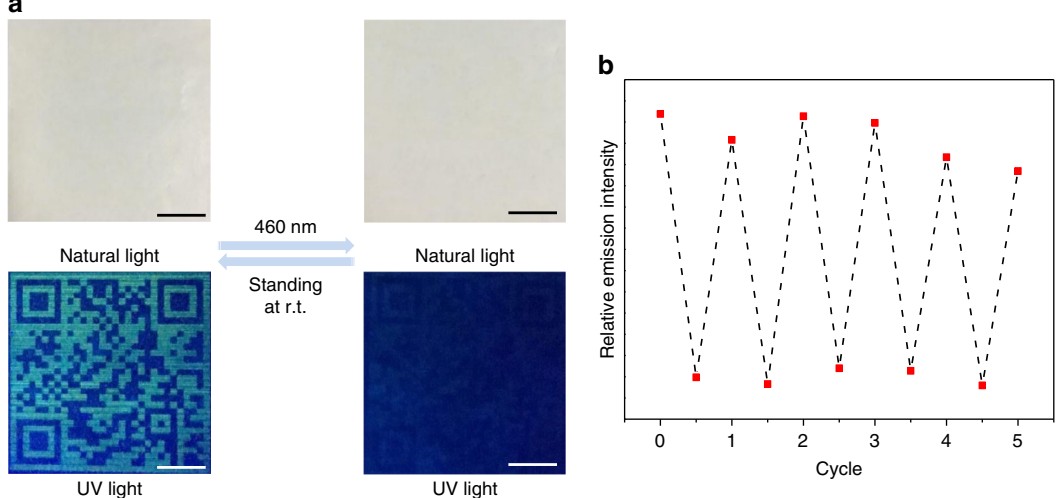

**Fig. 8** Single-mode anti-counterfeiting inks from **1**. **a** Photographs of the fluorescent QR code (size: 4 × 4 cm on paper) from **1** and its photo-responsiveness. Scale bar: 1 cm. **b** Relative fluorescence intensity ($\lambda = 512$ nm) for the QR code as a function of the cyclic numbers

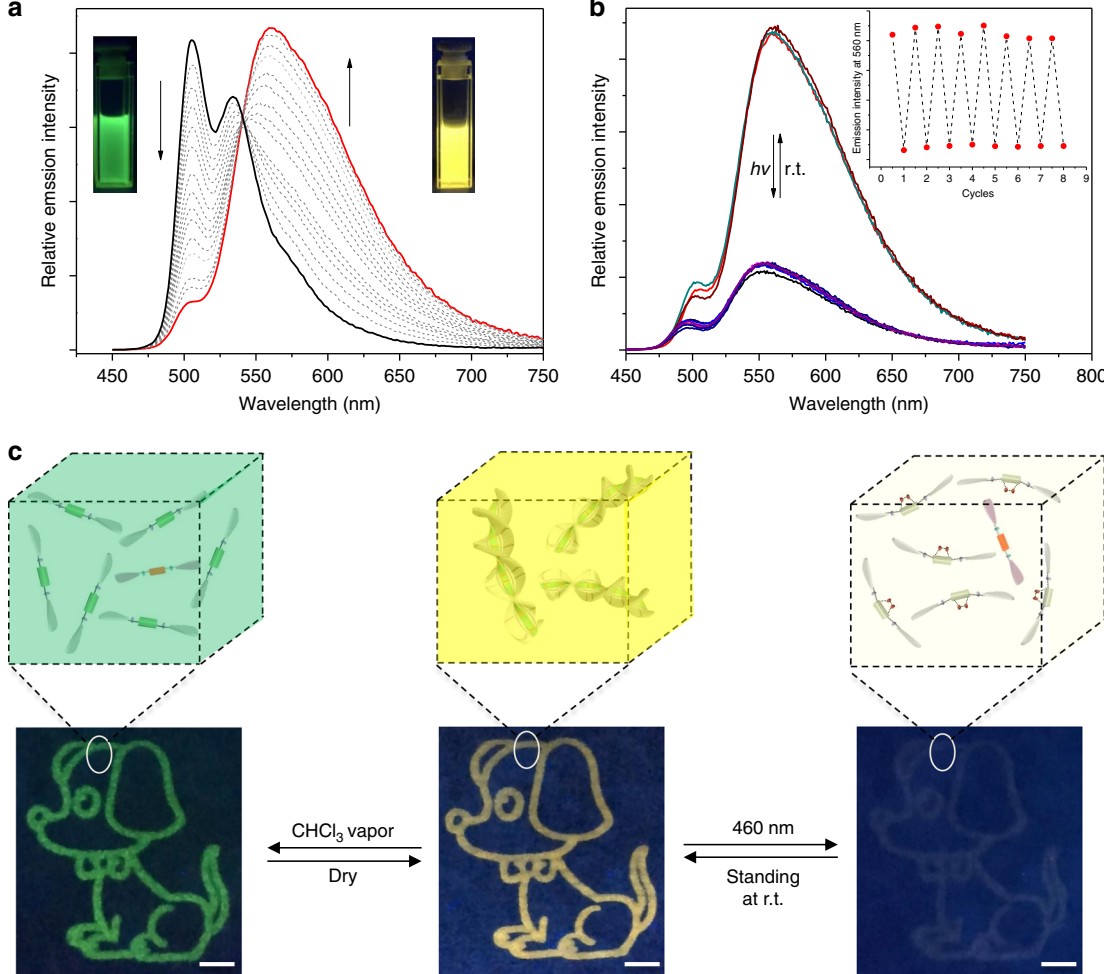

**Fig. 9** Dual-mode anti-counterfeiting inks from **1/2**. **a** Fluorescent spectra of **1** ($4.00 \times 10^{-5}$ M in MCH) via the gradual addition of **2** (0–10 mol%).
**b** Fluorescent spectra of **1/2** (10 mol% of **2**, $4.00 \times 10^{-5}$ M for **1**, $\lambda_{ex} = 425$ nm) upon successive 460 nm light irradiation and room-temperature standing. Inset: intensity changes at 560 nm. **c** Dual-stimuli responsive dog pattern by inkjet printing **1/2** ($4.00 \times 10^{-5}$ M of **1** in MCH, 2.5 mol% of **2**) on the weighting paper, together with the cartoon representation for the corresponding structures. The images are taken under 365 nm UV handhold lamp. Scale bars: 5 mm

(Supplementary Fig. 31). Overall, co-assembly **1/2** features three distinct emission signals (yellow, green, and weak fluorescence) upon light/vapor treatment, which is employed for dual-mode anti-counterfeiting applications with enhanced security reliability.

## Discussion

In summary, we have successfully developed photo-responsive supramolecular polymers on the basis of multicycle anthracene–endoperoxide switching. Due to Pt(II) $\sigma$-metalation effect, photo-oxygenation of **1** to **1**-$O_2$ can be proceeded via the mild visible light irradiation, while the backward conversion occurs spontaneously at room temperature. The fluorescent changes of **1** in response to light can be macroscopically amplified, because of the involvement of nucleation–elongation cooperative supramolecular polymerization process. For the resulting supramolecular copolymers **1/2**, FRET efficiency is orthogonally modulated via light and vapor stimuli, which facilitates the fabrication of dual-mode fluorescent anti-counterfeiting inks. Hence, supramolecular polymerization of photo-fluorochromic molecules provides an effective strategy toward high-performance anti-counterfeit materials with high security reliability, fast response, and ease of operation under the mild conditions.

## Methods

**Measurements**. $^1$H NMR spectra were collected on a Varian Unity INOVA-300 spectrometer with TMS as the internal standard. $^{13}$C NMR spectra were recorded on a Varian Unity INOVA-300 spectrometer at 75 MHz. MALDI-TOF measurements were recorded on a Bruker Autoflex Speed spectrometer with DCTB as the matrix. UV–Vis spectra were recorded on a UV-1800 Shimadzu spectrometer. CD measurements were performed on a Jasco J-1500 circular dichroism spectrometer, equipped with a PFD-425S/15 Peltier-type temperature controller. Solution excitation and steady-state fluorescence emission spectra were recorded on a FluoroMax-4 spectrofluorometer (Horiba Scientific) and analyzed with an Origin (v8.0) integrated software FluoroEssence (v2.2). TEM images were performed on a Tecnai G2 Spirit BioTWIN electron microscope (acceleration voltage: 120 kV). Electron paramagnetic resonance (EPR) measurements were performed on a JEOL JES-FA200 apparatus, with the utilization of 2,2,6,6-tetra-methylpiperidine as the spin trap for $^1O_2$. Fluorescence microscopy images were performed on an inverted fluorescence microscope (Olympus IX81).

**Theoretical calculations**. All optimized structures were performed on Gaussian 09 software packages. All of the non-metallic elements (C, H, O, N, and P) were described by PBEPBE/6-31G computational method, while Pt atoms were described by LANL2DZ core potential. There are no imagery frequencies for the optimized geometries.

**Inkjet printing experiments**. Printing tests were performed on an HP inkjet office printer (HP Deskjet 2131 model) with the customized HP803 black ink cartridges. The filled inks (black) from the inkjet cartridge were removed, and the cartridge was washed extensively with ethanol, water, and dried with $N_2$ blowing. The inks (0.5 mL) of **1** (1 mM) or **1/2** (2.5 mol% of **2**, 0.5 mM for **1**) were then loaded in the clean black ink cartridge to perform the printing experiments.

## Data availability

The data that support the findings of this study are available from the corresponding author on request.

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

## Acknowledgements

We acknowledge the funding support from National Natural Science Foundation of China (21871245 and 21674106), CAS Youth Innovation Promotion Association (2015365), and the Fundamental Research Funds for the Central Universities (WK3450000004).

## Author contributions

Z.G. and F.W. conceived the idea for this project. Z.G. performed the experiments, analyzed the data, and produced the artwork under the direction of F.W. Y.H. contributed to the theoretical calculations. All authors contributed to the manuscript preparation.

## Additional information

**Competing interests:** The authors declare no competing interests.

