## [Peer Review File · Nature Communications]

Reviewers' comments:

Reviewer #1 (Remarks to the Author):

This manuscript submitted by Wang and co-authors reported a novel type of photo-responsive supramolecular polymers for fluorescent anti-counterfeit applications by taking advantage of multicycle anthracene–endoperoxide switching properties. The photo-oxygenation of 1 to 1-O₂ can be proceeded via the mild visible light irradiation due to Pt(II) σ -metalation effect, while the backward conversion occurs spontaneously at room temperature. Because of the involvement of nucleation–elongation cooperative supramolecular polymerization process, the fluorescent changes of 1 in response to light can be macroscopically amplified. Finally, they used supramolecular copolymers 1/2 to prepare a dual-mode fluorescent anticounterfeiting inks based on FRET efficiency which can be orthogonally modulated via light- and vapor- stimuli. This approach is attractive and is potentially useful in anticounterfeit. Therefore, I recommend publication of the manuscript in *Nature Communications* after some concerns being addressed.

1. what is the advantage of this method compared to other reported methods?
2. In the printing experiment, organic solvent MCH must be used. So how about the safety of using of MCH.
3. It is suggested that the applicability of the method should be discussed in detail.
4. As a anti-counterfeiting material, the stability of materials is very important. So how is the stability and sustainability of the compounds 1/2?
5. why did the backward conversion occur at room temperature? The mechanism should be described.
6. In Fig. 3c, the mathematical fitting is not good. It should be re-fitted.

Reviewer #2 (Remarks to the Author):

The authors present an anti-counterfeiting technique realized by ink-jet printing of an anthracene-carrying supramolecular polymer doped with a FRET acceptor. The normally invisible image can emerge in either yellow or green color depending on the conditions.

The manuscript is well written and the results of each step are supported with strong evidences such as TEM, NMR, ESR, UV/vis, fluorescence and CD-spectroscopy.

A topic has been chosen, which is of great importance and interest for a very broad spectrum of chemists.

Although there exist numerous examples of anti-counterfeiting techniques, e. g. by using full-color inks (*Adv. Funct. Mater.* 2014, 24, 5029), triple mode (luminescence/upconversion/phosphorescence) emitting materials (*Angew. Chem.* 2016, 55, 7231) or holographic viewing angle depended techniques (*Nature Sci. Rep.* 6, 30885), the manuscript of the authors bears novel ideas, which puts this work at a very high level. A good example is the interruption of FRET between 1 and 2 by “fuming” to switch from yellow to green emission.

On the other hand, both phenomena (the photochromism of anthracenes and their FRET towards an acceptor) are extensively investigated in literature and come very close to this application (singlet-oxygen based photopatterning with acenes and interruption of FRET between acenes and energy-transfer-donors as for example used for oxygen sensors). Thus, my decision is a borderline case, but I would agree to accept the manuscript for a publication in *nature communications*.

At least some parts have to be modified prior to publication:

1. Line 184: The role of the Pt(PET₃)₂ moiety should be explained more clearly: The comparison of the irradiation experiments of 1 versus 5 shows that the reaction of a simple alkynylanthracene is

slower and ceases at a certain point. Describe why these decay curves look so different. Is this because the σ -metalation causes a more efficient population of T1 or a more sufficient energy transfer to $3O_2$? Or is it simply owed to the shift of the absorption wavelength? Is the recovery of 5 <100 %? Are then other products formed upon irradiation?

2. Line 195: When the kinetics in two different solvents ($CHCl_3$ / MCH) are compared, strong differences have to be expected owed to differences in polarizability (see Aubry, J. Am. Chem. Soc. 1995, 117, 9159). Thus, the differences do not arise exclusively from the supramolecular polymeric state. Therefore, the sentence in lines 201/202 should be modified.

3. Line 252: It should be mentioned that 2 still shows weak fluorescence in either the absence of 1 or with photooxygenated 1 upon irradiation with 365 nm.

4. Axis labeling in SI Figure S14b: Replace "irradiation time" by "time standing in the dark"

Reviewer #3 (Remarks to the Author):

Purpose of the paper is the development of an innovative counterfeiting technology which uses photofluorochromic compounds. Using a supramolecular approach is an interesting way of improving the quality (in terms of stability) of the ink. Also the possibility of achieving multimode response thanks to FRET is very appealing.

The data presented to prove the soundness of the declared objective are convincing and the characterization provided satisfactory.

I just have a minor remark: the authors prove the cyclability of the process, which is promising, but what can they say about the long term stability of the ink itself (when printed)? This is also an important issue in terms of applicability of the idea.

In conclusion I believe that the subject is of interest to a vast readership and can be published.

Reviewer #1:

1) What is the advantage of this method compared to other reported methods?

Response: The current anti-counterfeiting method, which display reversible emission outputs for the supramolecular polymers in response to external stimuli, is superior to the conventional methods due to the following points. i) **Excellent processability and ease of operation:** Ascribed to the involvement of cooperative (nucleation–elongation) supramolecular polymerization mechanism, both the self-assembled and co-assembled systems feature outstanding processability due to their high-molecular-weight character. As a consequence, the resulting supramolecular polymers (**1** or **1/2**) can be conveniently filled in the customized cartridge of an inkjet printer, which facilitates to print various types of anti-counterfeiting patterns. ii) **Rapid responsiveness under mild conditions:** The anti-counterfeiting property is dependent on the reversible “on-off” emission of the anthracene–endoperoxide transition processes. Due to Pt(II) σ -metalation effect, photo-oxygenation of anthracene **1** to endoperoxide **1-O₂** can be rapidly proceeded under the visible light irradiation conditions, while the backward conversion occurs spontaneously at room temperature. These mild and rapid anthracene–endoperoxide inter-conversion conditions benefit for the long-term use of anti-counterfeiting inks for multiple cycles. iii) **Enhanced security reliability:** Fluorescence resonance energy transfer (FRET) process takes place *via* two-component co-assembly strategy (complex **1/2**), which can be delicately regulated by either light-triggered anthracene–endoperoxide conversion or vapor-induced monomer-polymer transition. As a consequence, it leads to the reversible transformation among three distinct emission states with high-contrast signal changes. The dual-mode anti-counterfeiting patterns guarantees enhanced security reliability, as

compared to the single-mode counterparts. More importantly, the patterns feature the concealing capabilities, since the reversible changes are invisible under natural light, and can only be observed under UV light. In conclude, cooperative supramolecular polymerization of photo-fluorochromic molecules demonstrates the efficiency and superiority to develop high-performance anticounterfeit materials.

2) In the printing experiment, organic solvent MCH must be used. So how about the safety of using of MCH.

Response: In the current anti-counterfeiting materials, methylcyclohexane (MCH, boiling point: 101 °C) is employed as the solvent due to its compatibility to the side peripheries of the designed molecules. This apolar solvent is compatible to the cartridge of inkjet printer, and can be loaded for several days without any corrosion. It is totally evaporated during the ink-jet printing process. According to the previously reported data, this solvent is low toxicity to organism (LD_{50} : 2250 mg/kg for mouse, oral). Hence, we consider that the solvent can be utilized to demonstrate the new anti-counterfeiting concept on the basis of fluorescent supramolecular polymers. In the future studies, we can further modify the hydrophobic side chains by the hydrophilic poly-/oligo-(ethylene glycol) units in the designed monomers, which would be more environmentally friendly for anti-counterfeiting applications, since water can be employed as the inkjet printing solvent.

3) It is suggested that the applicability of the method should be discussed in detail.

Response: Based on the reviewer's comment, we have added more experiments to demonstrate the applicability of the current method. In particular, various types of non-fluorescent papers have been employed to inkjet-print the "dog" pattern, including the weighing paper, Xuan paper and the filter paper. All of them display yellow emission colors, and show the excellent fluorescent anti-counterfeiting behaviors (see Fig. R1a and Supplementary Fig. S30). Moreover, we have evaluated the waterproof capability of the printed pattern on the weighing paper. Upon immersing the inkjet-printing pattern into water overnight, the yellow fluorescence is still maintained (see Fig. R1b and Supplementary Fig. S30). In addition, we have also shown that the fluorescent inks can be directly loaded into the fountain pen, which facilitate to draw the pattern picture (see

Supplementary Fig. S32). Accordingly, the current method demonstrates the broad applicability toward high-performance anti-counterfeit materials.

Figure R1. a) Photographs of the photo-responsive fluorescent “dog” patterns printed on the weighing paper, Xuan paper, and the filter paper, respectively. b) Waterproof test for the printed pattern, by immersing the weighing paper into water overnight.

4) *As an anti-counterfeiting material, the stability of materials is very important. So how is the stability and sustainability of the compounds 1/2?*

Response: According to the reviewer’s comment, we have investigated the stability of the anti-counterfeiting inks derived from **1/2**, by placing the printed “dog” pattern under the ambient conditions for two weeks (from the reception of reviewer’s comment to the submission of the revised manuscript). As can be seen from Fig. R2a, the “dog” pattern maintains yellow fluorescence after two weeks, and almost no emission decay can be visualized under the UV lamp. On the basis of fluorescent measurements, it is also concluded that no obvious changes are found between the “fresh-prepared” and “aged” patterns (see Fig. R2b and Supplementary Fig. 31). Overall, the anti-counterfeiting fluorescent inks derived from **1/2** feature sufficient stability and sustainability.

Figure R2. a) Photographs and b) fluorescence spectra of the fluorescent “dog” patterns on day 1 and after 2 weeks.

5) Why did the backward conversion occur at room temperature? The mechanism should be described.

Response: For the traditional endoperoxide-to-anthracene deoxygenation reaction, harsh reaction conditions such as high-energy UV light and thermolysis are required (*J. Org. Chem.*, **2002**, 67, 916; *J. Am. Chem. Soc.*, **2007**, 129, 1488; *Dalton Trans.*, **2018**, 47, 2769). In the current system, deoxygenation of endoperoxide **1-O₂** to anthracene **1** can occur at room temperature. The mild reaction condition is primarily ascribed to the Pt(II) σ -metalation effect. According to the previous report (Schanze *et al.*, *J. Am. Chem. Soc.* **2008**, 130, 2535), there are two different conformations for the platinum(II) acetylide oligomers, originating from the carbon-carbon and carbon-platinum rotation. One possible conformation is the perpendicular orientation of the central anthracene ring with relative to the planes defined by the square planar PtP₂C₂ units, while another one is the coplanar orientation of the anthracene ring to the PtP₂C₂ planes. The former one is more favored at the ground state (as reflected by the DFT calculation of **1**), while the latter one can exist at the excited state due to the sterically constrained effect. In terms of **1-O₂**, the bulky endoperoxide ring imposes larger hindrance with respect to the neighboring two Pt(PEt₃)₂ units. To release the steric strain, it is capable of converting **1-O₂** to **1** at room temperature in a spontaneous manner. Based on the reviewer’s suggestion, we have added some explanation for the mild endoperoxide-to-anthracene conversion of **1-O₂** in the revised manuscript.

6) In Fig. 3c, the mathematical fitting is not good. It should be re-fitted.

Response: Based on the reviewer's suggestion, we have re-performed the mathematical fitting for the cooperative supramolecular polymerization of monomer **3**. The corresponding thermodynamic parameters display the slight changes, which are listed in Table 1 of the main text.

Reviewer #2:

1) Line 184: The role of the Pt(PEt₃)₂ moiety should be explained more clearly: The comparison of the irradiation experiments of **1** versus **5** shows that the reaction of a simple alkynylanthracene is slower and ceases at a certain point. Describe why these decay curves look so different. Is this because the σ -metalation causes a more efficient population of T1 or a more sufficient energy transfer to ³O₂? Or is it simply owed to the shift of the absorption wavelength? Is the recovery of **5** <100 %? Are then other products formed upon irradiation?

Response: For the designed monomer **1**, Pt(II) σ -metalation of 9,10-dialkynyl anthracene offers the following advantages. First, it allows fine-tuning of the HOMO-LUMO gap through the interaction of the metal *d*-orbitals with the ligand *p*-orbitals, leading to the enhancement of π -electron delocalization and red-shifting of ¹L_a absorption band. As a result, photo-oxygenation of **1** can be proceeded *via* the less destructive visible light source (460 nm LED lamp). Second, the efficient inter-system crossing enhances the spin-orbit coupling, thus facilitates the population of the triplet state. The latter effect can be evidenced by electron paramagnetic resonance (EPR) measurements, which show the capture of singlet oxygen (¹O₂) generated *in situ* by TEMP to form the TEMPO radical. Hence, the anthracene-to-endoperoxide photo-switching process is faster and more thorough for **1** than that of **5**. Based on the reviewer's comment, we have added more sentences to describe the Pt(II) σ -metalation effect in the revised manuscript.

Moreover, we have further examined the photo-conversion capability of **5** on the basis of ¹H NMR experiments. Upon photo-irradiation (460 nm LED lamp, 12 W) of **5** (293 K, in CDCl₃) for an hour, only 60% conversion takes place for the photo-oxygenation process. Complete recovery of **5** from **5**-O₂ can be achieved upon heating. During the photo-irradiation process, the sample only contains **5** and **5**-O₂, while no other products can be detected (see Fig. R3 and Supplementary Fig. 14).

Figure R3. Partial ^1H NMR spectra (CDCl_3 , 298 K) of **5**: a) before irradiation, b) after irradiation for 1 hour, and c) heating at 50°C for 15 min.

2) Line 195: When the kinetics in two different solvents (CHCl_3/MCH) are compared, strong differences have to be expected owed to differences in polarizability (see Aubry, *J. Am. Chem. Soc.* 1995, 117, 9159). Thus, the differences do not arise exclusively from the supramolecular polymeric state. Therefore, the sentence in lines 201/202 should be modified.

Response: Based on the reviewer's comment, we have added the solvent polarity effect, which also influences the kinetics of anthracene–endoperoxide photo-switching process.

3) Line 252: It should be mentioned that **2** still shows weak fluorescence in either the absence of **1** or with photo-oxygenated **1** upon irradiation with 365 nm.

Response: Based on the reviewer's suggestion, we have modified the description in the revised manuscript, by replacing “weak fluorescence” instead of “non-fluorescence” in the revised manuscript.

4) Axis labeling in SI Figure S14b: Replace “irradiation time” by “time standing in the dark”

Response: We have revised the error in the mentioned place.

Reviewer #3:

I just have a minor remark: the authors prove the cyclability of the process, which is promising, but what can they say about the long-term stability of the ink itself (when printed)? This is also an important issue in terms of applicability of the idea.

Response: The comment is similar to the forth comment made by reviewer 1. We have shown that the inkjet-printing dog pattern derived from **1/2** is stable under ambient conditions for two weeks (see Fig. R2 and Supplementary Fig. 31), which is the time duration from the reception of reviewer's comment to the submission of the revised manuscript. Therefore, it can be concluded that the anti-counterfeiting fluorescent inks feature sufficient stability and durability.

Some of the data, results, and discussion are added in the main text and supporting information (see the revised manuscript with red highlights). We are hopeful that it now meets your standards for acceptance. Thanks!

REVIEWERS' COMMENTS:

Reviewer #1 (Remarks to the Author):

Authors have addressed all issues carefully. Therefore, I recommend this revised manuscript publication in Nature Communications.

Reviewer #2 (Remarks to the Author):

The four points I suggested for revision, have been satisfyingly addressed and modifications are properly placed in the new revised version. There is now no more reason to withhold the publication.

Reviewer #3 (Remarks to the Author):

In the revised version of the manuscript, the authors have satisfactorily answered all the questions. In my opinion, the paper can be published in its present form.